# Dysregulated Hemostasis and Immunothrombosis in Cerebral Cavernous Malformations

**DOI:** 10.3390/ijms232012575

**Published:** 2022-10-20

**Authors:** Maria Ascencion Globisch, Favour Chinyere Onyeogaziri, Ross Osborne Smith, Maximiliano Arce, Peetra Ulrica Magnusson

**Affiliations:** Department of Immunology, Genetics and Pathology, Uppsala University, 751 85 Uppsala, Sweden

**Keywords:** BBB, bleeding, CCM, coagulation, endothelium, immunothrombosis, NETosis, neuroinflammation, hemostasis, hypoxia

## Abstract

Cerebral cavernous malformation (CCM) is a neurovascular disease that affects 0.5% of the general population. For a long time, CCM research focused on genetic mutations, endothelial junctions and proliferation, but recently, transcriptome and proteome studies have revealed that the hemostatic system and neuroinflammation play a crucial role in the development and severity of cavernomas, with some of these publications coming from our group. The aim of this review is to give an overview of the latest molecular insights into the interaction between CCM-deficient endothelial cells with blood components and the neurovascular unit. Specifically, we underscore how endothelial dysfunction can result in dysregulated hemostasis, bleeding, hypoxia and neurological symptoms. We conducted a thorough review of the literature and found a field that is increasingly poised to regard CCM as a hemostatic disease, which may have implications for therapy.

## 1. Introduction

Cerebral cavernous malformation (CCM) is a disease that affects the vasculature of the central nervous system (CNS). Cavernous malformations (or cavernomas) are fragile, leaky, mulberry-like lesions with abnormal blood flow. CCM patients may exhibit symptoms such as epileptic seizures, focal neurological deficits, intracranial bleeding and hemorrhagic strokes. Surgical resection is the only treatment for CCM patients; however, propranolol, atorvastatin and superoxide dismutase are currently being evaluated in phase-2 clinical trials [1].

Cavernomas appear sporadically (1:200) or as an autosomal dominant (1:10,000) loss-of-function germline mutation of *KRIT1* (*CCM1*), *Malcavernin (CCM2)*, or *PDCD10* (*CCM3*) in endothelial cells of the CNS [2,3]. Cavernomas develop via clonal expansion of mutated endothelial cells [4,5] and recently, gain-of-function mutations in *PIK3CA* were also identified in CCM patient biopsies [6]. Furthermore, the acquisition of somatic mutations and exposure to systemic proinflammatory molecules can fuel cavernoma formation and worsen already existing lesions [7]. Therefore, it is evident that CCM proteins are crucial regulators of endothelial quiescence and plasticity in the CNS. The vast phenotypic plasticity of endothelial cells is a direct reflection of their exposure to the environment. A study by Maddaluno et al. showed that murine endothelial cells lacking CCM1 undergo an endothelial-to-mesenchymal transition (EndMT) [8] a process which, in CCM, depends on the transcriptional activity of Krüppel-like factor 4 (KLF4) [9,10]. During EndMT, endothelial cells acquire a mesenchymal phenotype and become thrombogenic, proliferative and highly mobile [11] suggesting that this process could drive lesion progression and instability.

Within the past years, several studies have contributed to our understanding of how the loss of CCM proteins triggers lesion development, however, the exact mechanisms for how bleedings occur and how they could lead to neurological symptoms remain elusive. For many years the leaky and fragile nature of endothelial junctions was thought to be the main factor underlying intracranial bleeding. Nevertheless, transcriptome and proteome studies have revealed that the hemostatic system and inflammation play crucial roles in the development and severity of cavernomas [12,13,14,15,16,17]. Notably, in 2019, Lopez-Ramirez et al. reported that the anticoagulant properties of endothelial cells in CCM promoted intracranial hemorrhage [12]. Our group has also worked on this topic extensively and has recently reported that individual CCM lesions can contain both anticoagulant and procoagulant regions that contribute to hemorrhage, thrombi and cerebral hypoxia [17]. These studies raise the possibility that CCM, at least to some extent, can be termed a hemostatic disease. In this review, we present evidence of the interaction between CCM-deficient endothelial cells, blood components and elements of the neurovascular unit. We also propose that cavernomas have a distinct microenvironment that may give insights into novel therapeutics for patients who suffer from CCM.

## 2. Endothelial Dysfunction in CCM

The most severe pathological events in CCM are episodes of cerebral bleeding and hemorrhagic stroke. These symptoms have commonly been associated with a defective endothelial barrier caused by a loss of tight and adherent junctions. In endothelial cells, CCM proteins are important modulators of junctional and cytoskeletal stability. The absence of these proteins results in the disruption of vascular endothelial cadherin (VE-cadherin), an endothelial-specific protein that regulates cell-to-cell junctions, cell polarity and permeability [18]. Notably, the activation of Ras homolog family member A (RhoA) and its effector Rho effector kinase (ROCK) modulate VE-cadherin disruption in CCM [19]. Mechanistically, RhoA and ROCK modulate endothelial permeability, by the phosphorylation of myosin light chain (pMLC), which in turn initiates the contraction of F-actin filaments that firmly attach to adherens junctional proteins, like VE-cadherin [20]. The increasing tension, due to the contraction and stabilization of F-actin radial stress fibers, results in the formation of gaps between cells and the disruption of cell-to-cell contacts [20]. Interestingly, in CCM, the pharmacological inhibition of ROCK reverses endothelial hyperpermeability and leakage in vitro and reduces bleeding in vivo [19,21]. Importantly, the RhoA pathway has been proposed to play a central role in the pathophysiology of CCM, as pMLC is increased in endothelial cells of sporadic and familial CCM patients [19]. Consequently, the overactivation of the RhoA pathway could be an important factor contributing to endothelial dysfunction and, therefore, affecting the hemostatic properties of an essential gatekeeper of hemostasis.

## 3. The Hemostatic System in CCM Is Dysregulated

### 3.1. The Hemostatic System

The hemostatic system consists of the endothelium, platelets, coagulation factors and proteins that regulate fibrinolysis [22,23]. Under physiological conditions, these components work together to maintain hemostatic balance by halting hemorrhage without completely occluding the vessel with thrombi [23]. Upon endothelial damage, platelets become activated and adhere to the site of injury to form a platelet plug; a process termed *primary hemostasis* [22]. Damaged endothelial cells and activated platelets then release procoagulant molecules that promote the proteolytical cleavage and activation of coagulation factors to initiate *secondary hemostasis*. *Secondary hemostasis* is divided into two pathways that lead to the formation of a fibrin clot. The first pathway is the intrinsic pathway (or contact activation pathway) which involves high molecular weight kininogen, prekallikrein and coagulation factor XII that when in contact with an artificial or biological negatively charged surface (such as kaolin or a thrombus), trigger the blood coagulation cascade [24]. The second pathway is the extrinsic pathway (or tissue factor (TF) pathway). Tissue factor is expressed by many cells in the brain including astrocytes, neurons and oligodendrocytes [25,26], and when in contact with blood, it strongly binds circulating FVII to create a complex and trigger a catalytic event that results in the formation of activated FVII (FVIIa) [24]. Both the intrinsic and extrinsic pathways converge into the common pathway, where activated FX (FXa) is formed, cleaving circulating prothrombin and leading to the formation of thrombin. Then, the proteolytic activity of thrombin allows the cleavage of fibrinogen into fibrin monomers, which are used to turn the platelet plug into a stable cross-linked fibrin clot [24]. As a result of the activated coagulation cascade, the endothelium releases tissue plasminogen activator (tPA). Together with urokinase plasminogen activator (uPA), tPA binds circulating plasminogen and coverts it to plasmin, the main proteolytic enzyme involved in fibrinolysis (*tertiary hemostasis*). The role of plasmin is to cleave fibrin into fibrin degradation products and dissolve the clot [27].

### 3.2. Are vWF-Activated Platelets Supporting the Activation of the Intrinsic Pathway in CCM?

The vasculature is a natural and dynamic modulator of hemostasis. Under physiological conditions, endothelial cells form a nonadherent surface that precludes platelet adhesion and prevents the activation of the coagulation cascade [28]. Alterations in vascular homeostasis can lead to a physiological response in which the endothelium switches from an anticoagulant to a prothrombotic state. One of the earliest phenotypic modifications is the exposure and release of von Willebrand factor (vWF) from injured endothelial cells. Von Willebrand factor is a glycoprotein that is produced by endothelial cells and is stored in Weibel-Palade bodies, structures that were discovered by Ewald R. Weibel and George E. Palade in 1964 [29]. Upon stimulation (e.g., by thrombin), vWF is quickly released and exposed to the luminal side of the damaged endothelium [20]. Exposed, and uncoiled, vWF mediates platelet adhesion through its interaction with the glycoprotein (GP)Ib-IX-V complex which is located on the surface of circulating platelets [30]. Platelet adhesion and activation is supported by the binding of GPIb-IX-V to vWF. This creates a positive feedback loop that promotes the release of vWF from activated platelets. Interestingly, vWF accumulates in the lumen of human cavernomas, and in vitro models of CCM show that vWF increases in endothelial cells and forms vWF-strings in the absence of CCM1 or CCM3 [17,31]. These data suggest that vWF might be released directly after CCM genes are inactivated; therefore, we hypothesize that CCM-deficient endothelial cells recruit and activate platelets in the early phases of lesion development. In turn, activated platelets release polyphosphates [32] which could activate circulating factor XII and initiate the intrinsic pathway of the blood coagulation cascade [33]. However, more recent studies showed that platelet-derived polyphosphates are weak activators of FXII, therefore, they are not physiologically relevant activators of this blood protease [34,35]. Overall, it seems that the intrinsic pathway could be one of the first pathways to be activated during blood coagulation in CCM, as dysfunctional endothelial cells expose vWF, recruiting and activating platelets which in turn may activate factor XII. The exact mechanisms that result in vWF rearrangement, activated platelets and blood clotting in CCM, remains elusive. Indeed, the overactivation of the RhoA-ROCK pathway (due to the loss of the CCM proteins) could partially explain this phenomenon, as this pathway modulates thrombin-induced vWF release from endothelial cells [36].

### 3.3. Is the Extrinsic Pathway Activated When CCM Lesions Are Established?

Tissue factor, a key molecule involved in the activation of the extrinsic pathway of the coagulation cascade, is expressed by parenchymal cells surrounding the lesions of CCM3-null mice [17]. When TF comes in contact with blood, it initiates the coagulation cascade by binding and activating coagulation factor VIIa (FVII/FVIIa). The TF-FVIIa complex promotes the activation of coagulation factor X (FXa) and unleashes a proteolytic cascade that converts prothrombin into thrombin, fibrinogen to fibrin to eventually form a blood clot. Noshiro et al. show that increased transcript levels of TF correlated with hemorrhagic events in CCM patients [37]. These data suggest that the TF pathway of secondary hemostasis is somehow related to bleeding in CCM.

### 3.4. CCM Lesions Have Stable Thrombi with Polyhedrocytes

Although CCM is considered a bleeding disease, thrombi have been reported in surgically resected cavernomas [38,39,40]. However, due to the challenges of acquiring patient material, much of the in vivo research presented in this review is derived from genetically induced mouse models of CCM. It should be acknowledged that, while these strains have been undeniably useful, there are specific caveats to keep in mind. Namely, that lesion formation in mice occurs over a period of days to months and in many of the models, lesions are restricted to the cerebellum, whereas in CCM patients, lesions can develop over months and years and are spread throughout the CNS. Furthermore, acute hemorrhage, which is a serious symptom in patients, is rarely seen with the commonly used mouse strains. With regards to thrombi observed in CCM mouse models, a recent article identified fibrin clots with fibronectin, vWF (Table 1) and activated platelets [17]. These data suggest that blood clots in CCM are stable, as blood glycoproteins, such as fibronectin and vWF, can stabilize blood clots in vivo [41,42,43].

In addition, polyhedrocytes are also seen in both murine and human cavernomas [17] (Table 2). Polyhedrocytes are red blood cells (RBCs) that become tightly compressed during clot contraction where activated contractile platelets pull fibrin strands to reduce the clot size and restore blood flow [44]. However, not all RBCs within clots become polyhedral during clot contraction. Interestingly, a study by Chernysh et al. compared clot compositions between arteries and veins and shows that venous thrombi consist of more polyhedrocytes compared to arterial thrombi [45]. Since CCM lesions originate from venous endothelial cells [15], they may be more prone to polyhedrocyte formation during a clot contraction. Clot formation in veins has been attributed to hypercoagulability and reduced blood flow (Virchow’s triad) [46] all of which, as discussed in prior sections, occur in CCM. Under low shear stress, RBCs aggregate into a rouleaux and pile along the axis of flow, increasing the blood viscosity and margination of platelets [47]. It is conceivable that the already reduced blood flow in cavernomas is further stagnated by the formation of clots, creating an environment that further promotes clotting. We postulate that low shear stress in CCM lesions promotes the aggregation of RBCs into a rouleaux which makes them more susceptible to become polyhedral during clot contraction. Contracted clots with polyhedrocytes might have dual consequences; first, they may reduce the risk of hemorrhage since contracted clots are more compact and thus, are less prone to rupture than non-contracted clots [48]. Second, contracted clots, especially in veins, have been shown to be impermeable and could thus prevent fibrinolysis and hinder the access of therapeutic agents into the brain [44,45]. Importantly, the more a clot contracts, the more polyhedrocytes are formed [48]. Interestingly, the presence of polyhedrocytes in familial CCM human patient biopsies was reported to be inversely correlated with the expression of thrombomodulin in lesion endothelial cells [17] suggesting that the presence of polyhedrocytes may alter the hemostatic properties of endothelial cells, or vice versa. However, the clinical relevance of polyhedrocytes and stable clots in CCM remains to be explored.

### 3.5. Tertiary Hemostasis May Be Disrupted in CCM

The degradation of cross-linked fibrin is essential for the resolution of blood clots. The proteolytical activity of uPA and tPA allows the cleavage of circulating plasminogen and the formation of plasmin, thus promoting fibrinolysis. Plasminogen activator inhibitor-1 (PAI-1) is a protein that limits fibrinolysis by inhibiting the enzymatic activity of uPA and tPA, thus restricting plasmin generation [51,52]. PAI-1 is produced by endothelial cells, and interestingly, PAI-1 protein levels are highly upregulated in *CCM3* knock-down human brain endothelial cells. Moreover, PAI-1 transcript levels were significantly higher in cerebellar lesion endothelial cells of CCM mice [17]. Therefore, increased levels of PAI-1 could contribute to inefficient fibrinolysis in CCM lesions, resulting in long-lasting blood clots. Stable clots can undergo morphological modifications that can interfere with their dissolution, as strong clot contraction decreases the access of fibrinolytic proteins and impairs fibrinolysis [53]. The presence of polyhedrocytes in CCM clots (discussed earlier) indicates that thrombi within lesions undergo great contraction and, thus, may not be lysed efficiently. This observation suggests that tertiary hemostasis in CCM might be compromised. However, further studies are needed in this area to fully elucidate the molecular mechanisms of fibrinolysis in CCM.

### 3.6. Cavernomas Have Anticoagulant Regions

One of the most devastating side effects of cavernomas is intraparenchymal bleeding which may result in neurological symptoms such as migraines, focal neurological deficit and seizures. Bleeding cavernomas are associated with the loss of barrier function in brain endothelial cells [54]. However, in 2019, Lopez-Ramirez and colleagues shifted the perspective and investigated the anticoagulant properties of endothelial cells in CCM pathology [12]. They found that endothelial cells in murine models of CCM have elevated levels of thrombomodulin, activated protein C (APC, Table 1) and endothelial protein C receptor (EPCR) [12]. Thrombomodulin is expressed on endothelial cells and acts as a cofactor for thrombin during anticoagulation [55,56]. Thrombomodulin binds thrombin, inhibiting its interaction with procoagulant substrates, and instead promotes the activation of protein C to APC [57,58]. Importantly, Lopez-Ramirez and colleagues show that these proteins contribute to the bleeding phenotype in CCM. They also found that human CCM patients have increased levels of soluble thrombomodulin [12]. These data suggest that the hemostatic properties of the endothelium also play a role in hemorrhage and that blood glycoproteins may be used as biomarkers for hemorrhagic risk [12].

The overexpression of KLF2 and KLF4 is a hallmark of CCM [9,10], and these transcription factors can upregulate thrombomodulin and endothelial nitic oxide synthase (eNOS) [59,60]. Indeed, the upregulation of thrombomodulin and eNOS in CCM is KLF4 dependent [12]. Concomitantly, CCM3-deficient mouse brain endothelial cells produce more nitric oxide (NO) compared to control cells [12]. NO is a fast and potent vasodilator, but can also be considered an antithrombotic molecule, due to its inhibitory effect on platelet aggregation and activation [61]. Therefore, prolonged NO release in CCM lesions could exert an antithrombotic effect due to platelet inhibition and, ultimately, increase the probability of bleeding events.

Exposed anionic phospholipids, resulting from endothelial damage, provide a binding surface for the assembly of coagulation factors; however, these surfaces are shielded by annexin V, dampening coagulation [62,63]. Notably, annexin V was increased in a murine model of CCM [17]. The juxtaposition of the anticoagulant proteins thrombomodulin and annexin V, which are expressed in lesions with clots, suggests that the compensatory upregulation of the anticoagulant pathway in CCM is not sufficient to dampen the dysregulated hemostatic pathway. The inability of thrombomodulin to rescue hypercoagulation in CCM might be linked to the vascular growth factor angiopoietin-2 (Ang2). In vitro studies by Christopher Daly et al. show that Ang2 can bind thrombomodulin and inhibit its anticoagulant function [64]. The mechanism whereby Ang2 binds thrombomodulin and modulates coagulation is also observed in critically ill COVID-19 patients [65]. Thrombin induces the expression and exocytosis of Ang2 [66] and in CCM, there is increased exocytosis of Ang2 [67], which contributes to vascular dysfunction. It is thus probable that increased Ang2 in CCM inhibits thrombomodulin’s anticoagulant function and may contribute to clotting.

## 4. Hemodynamics and Hypoxia in CCM

Abrupt changes in blood flow due to malformations in vessel architecture can result in blood stasis [68]. A tortuous vascular phenotype is present in CCM lesions, which is driven by the shape of the characteristic mulberry-like vessels. It is known that the pathological dilations found in aneurysms can lead to turbulent, low-blood flow which promotes endothelial dysfunction, inflammation and even ruptures [69]. Likewise, CCM lesions are prone to blood stasis and disturbed flow due to their irregular shape. Vessel enlargement and disorganization could affect blood flow directionality, thus changing wall shear stress and leading to endothelial dysfunction (Figure 1).

Turbulent flow and low shear stress promote the activation of developmental signaling pathways, including the BMP, WNT, Notch and YAP-TAZ pathways [70]. Interestingly, clinical observations from a Hispanic cohort of CCM1 patients show that for every ten units increase in systolic blood pressure, lesion count decreased by 16% [71]. Moreover, no significant association with lesion count was observed for hypertension, suggesting that slightly higher systolic blood pressure, and potentially higher blood flow/shear stress, could have a beneficial effect on CCM pathogenesis. Strikingly, a recent study shows that blood flow suppresses vascular anomalies in a zebrafish model of CCM with a mutated *Krit1* gene in endothelial cells, but not in endocardial cells [72]. These results highlight the importance of blood flow as a contributing factor to vessel dilatation and enlargement in CCM, suggesting that physiological flow is a protective factor in this pathology. Notably, when CCM2-deficient endothelial cells are cultured in vitro and exposed to low shear stress, they upregulate several genes related to the WNT and transforming growth factor-β (TGF-β) pathways [73], which are both involved in the induction of EndMT in CCM [8]. Taken together, long-lasting clots within tortuous lesions present a scenario ripe for the generation of low shear stress and turbulent flow (Figure 1). Nevertheless, additional in vivo studies are needed to understand how disturbed flow (due to blood clots or vessel tortuosity) is impacting endothelial cells within CCM lesions.

Changes in blood supply and flow patterns are common after the coagulation cascade is triggered, and a blood clot is established. A blood clot may obstruct the delivery of oxygen to parenchymal cells which, consequently, could up-regulate pro-angiogenic and tissue remodeling factors, as a way to balance tissue homeostasis. Under physiological conditions, clots are resolved by the fibrinolytic machinery that is triggered once the initial damage has been addressed, ultimately resulting in the restoration of blood flow. However, the existence of established clots in CCM, characterized by the presence of polyhedrocytes (see above) [17], supports the premise that some CCM lesions are exposed to long-term partial occlusions. In addition, hypoxia surrounds the lesions of CCM-null mice and the presence of clots correlates with more hypoxia [17]. Therefore, ischemia and chronic hypoxia could be present within CCM lesions, accelerating and/or exacerbating different pathological aspects of the disease.

## 5. Are Platelets Enhancing EndMT in CCM-Deficient Endothelial Cells?

Endothelial cells undergo EndMT during the establishment and progression of cavernomas; this was experimentally demonstrated by Maddaluno and collaborators when they showed that the activation of TGF-β and BMP6 signaling pathways is crucial in the etiology of CCM [8]. Throughout the EndMT process, endothelial cells reduce the expression of essential proteins involved in the formation of the endothelial barrier, such as VE-cadherin and claudin-5 [74]. This suggests that EndMT is a process that contributes to endothelial barrier instability, and one that could intensify cavernoma hemorrhage. In CCM, a “wound” microenvironment is present, characterized by endothelial hyperproliferation, vessel remodeling and blood leakage. Platelets are central cellular components of the wound healing and tissue regeneration processes, and they are attracted to damaged and inflamed areas. Once activated by different stimuli (such as thrombin, vWF and collagen), platelets release their pro-angiogenic-TGF-ß-rich granules [75] within the injured area to promote blood clot formation, cell regeneration and wound healing [76]. Moreover, megakaryocytes and platelets contain BMP-6, a protein that both promotes EndMT and has been linked to CCM progression [8]. Despite the increasing number of studies linking platelets to specific pathologies, the role of stimulated-platelets (that can release TGF-β and BMP-6) in EndMT and CCM remains unclear.

## 6. Immunothrombosis and Neuroinflammation in CCM

### 6.1. The Role of PARs in CCM

Besides their role as effector proteins of the clotting cascade, FXa and thrombin can cleave the protease-activated receptors (PARs) [77]. The PARs are G protein-coupled receptors that are activated by a proteolytical cleavage on their extracellular portion, a process that generates a short peptide that works as an auto-ligand [78]. Protease-activated receptor-1 (PAR-1) is the most studied of the four PARs, and it is predominantly expressed by endothelial cells and platelets. In endothelial cells, both thrombin and FXa (with less affinity) cleave PAR-1 at the Arg41 residue and trigger a fast response that leads to junction dismantling and barrier disruption; a process that is mediated by RhoA [79]. In addition, FXa and thrombin promote long-term inflammation in endothelial cells, a process that is supported by IL-6 and IL-8 secretion as well as the luminal expression of ICAM-1 and VCAM-1 [80]. Interestingly, increased transcript levels of ICAM-1 and VCAM-1 have been reported in preclinical models of CCM [16].

PAR-1 has biased signaling in endothelial cells with different signaling outcomes that depend on the cleavage of specific amino acid residues within the extracellular domain [79]. A clear example of this biased signaling is the endothelial-protective signaling triggered by APC, a protease that is activated by the proteolytic cleavage of protein C by thrombin during thrombus formation. Unlike thrombin, APC cleaves PAR-1 at the Arg46 residue and promotes a cytoprotective effect within endothelial cells and increases the stability of the endothelial barrier [79]. Furthermore, APC treatment decreases leukocyte adhesion and transmigration in cremaster muscle venules during trauma and TNF-induced inflammation [81]. Interestingly, PAR-1 activation by APC increases the activity of Rac-1, a small GTPase that promotes adherens junction formation and barrier stabilization [82]. The promotion of Rac-1 activity counteracts the RhoA signaling pathway (overactivated in CCM lesions), suggesting that APC has an endothelial-protective effect through RhoA inhibition. This is exemplified in endothelial cells, where APC activates PAR-1 and suppresses the hyperpermeability triggered by thrombin [83], suggesting that both the Rac-1 and RhoA pathways are interacting and competing. Taken together, APC signaling through PAR-1 could stabilize the endothelial barrier by counteracting RhoA signaling and by decreasing inflammation. As a consequence of its cytoprotective effects on endothelial cells, APC could also fuel endothelial proliferation within CCM lesions. A recent study showed that APC-PAR-1 signaling promotes Akt phosphorylation at Ser473 through S1PR1 transactivation, thereby stimulating cell survival [84]. Similarly, the overexpression of KLF4 (see above) leads to Akt phosphorylation at Ser473 in cultured vein endothelial cells [9,10]. KLF4 gain-of-function in endothelial cells induces CCM-like brain lesions, wherein Rapamycin (inhibitor of the mTOR pathway, downstream of Akt) can prevent lesion formation [6]. Moreover, Uchiba and colleagues demonstrate that APC promotes MEK 1/2 and ERK 1/2 phosphorylation and leads to DNA synthesis and endothelial proliferation, events which were blocked with a neutralizing antibody targeting EPCR [85]. Further in vivo experiments showed that the topical administration of APC to mouse cornea promotes angiogenesis only in controls, but not in eNOS knock-out mice, suggesting that eNOS activation and NO production could be the link between APC activity and endothelial proliferation [85]. 

When considering that CCM lesions have procoagulant domains, it is plausible that PAR-1 activity, through FXa and thrombin cleavage, is worsening the already disrupted endothelial barrier and supporting inflammation. On the other hand, the physiological anticoagulant response through APC generation could activate PAR-1, triggering endothelial proliferation and angiogenesis, and thus, fueling cavernoma growth.

The protease-activated receptor 2 (PAR-2) is expressed by endothelial cells, and its activation has been linked to inflammation [86]. Interestingly, several proteases that are part of the hemostatic system (like FXa, kallikrein and plasmin) can cleave and activate PAR-2 [78]. Significantly, PAR-2 stimulation, either by FXa or specific agonist peptides, reduces vascular permeability in vitro, suggesting that PAR-2 downstream signaling could somehow exert a protective effect [87,88]. Nevertheless, the most recent reports suggest that FXa cleavage of PAR-2 results in increased permeability across an endothelial monolayer [89]. On the other hand, PAR-2 activation promotes endothelial cell proliferation in vitro [90] and retinal angiogenesis in vivo [91]. Interestingly, PAR-2 mRNA levels are highly upregulated in mouse brain endothelial cells isolated from CCM3-deficient mice [17]. Overall, further studies are needed to understand the importance of PAR signaling in CCM and the possible consequences in endothelial barrier dysfunction, immunothrombosis and lesion progression.

### 6.2. Immune Cells and Cytokines in CCM

The involvement of inflammation in CCM pathology is well established. The presence of different immune cell subtypes, from both the innate and adaptive immune responses, has been shown in both clinical and preclinical models of CCM [16,17,44,45] (Table 2). Neutrophils are the most abundant leukocytes in human blood and are one of the first cells recruited to an inflamed site [92]. Neutrophils play an important role in tissue repair and they are uniquely capable of producing neutrophil extracellular traps (NETs), DNA structures released upon chromatin decondensation [92]. Yau and colleagues showed that NETs are abundant in both human and preclinical models of CCM and that coagulation precedes NETosis [16]. NETs also serve as a scaffold for platelets, complement proteins and immune cells, possibly creating pocket-sites of chronic and continuous inflammation [93]. In preclinical models of CCM, platelets are activated and interact with NETs and polyhedrocytes [16,17] (Table 2). During NETosis, neutrophils release histones and myeloperoxidase which leads to endothelial cytotoxicity [94] that may result in worsening patient symptoms. Furthermore, it is conceivable that the cellular toxicity caused by NETosis products, such as cell-free DNA (cfDNA) and reactive oxygen species, could affect other cell types in the brain parenchyma. Release of cfDNA from neutrophils, or even from damaged endothelial cells, can trigger the activation of NFκB and MAPK pathways in endothelial cells and lead to the upregulation of adhesion molecules to propagate inflammation [95]. Notably, treating CCM-bearing mice with DNaseI to digest cfDNA decreases fibrinogen/fibrin deposition and immunoglobulin-G leakage into the brain parenchyma [16]. These data suggest that NETs have a role in promoting damage in the CCM microenvironment, however, the complete role of NETs in CCM is not yet known. Interestingly, the depletion of B-cells in CCM reduces lesion size but does not affect lesion initiation, indicating that the role of immune cells in CCM might primarily be in lesion progression [50].

Inflammatory chemokines and cytokines, such as CD14, IL-10, IL-6, IL-1β and soluble ROBO4, among others, are elevated in murine models of CCM and patients with cavernomas [16,37,96,97,98] (Table 3). These mediators are linked to different symptoms, such as bleeding and hemorrhages in CCM patients [37,96,97,98] justifying their potential role as biomarkers for disease severity. Of note, the reports on the cytokine IL-6 and hemorrhage in patients have been conflicting. Noshiro et al. showed an association between high IL-6 mRNA levels and an increased number of hemorrhage events in sporadic CCM patient samples [37]. However, in 2018, Girard et al., reported that reduced IL-6 levels in plasma from sporadic and familial patients were instead associated with hemorrhages [98]. This highlights how data obtained from in situ methods might differ from those obtained from plasma. The presence of clots within vessels could be the reason for this discrepancy, since cytokines and other blood proteins can get stuck within clots and thus, give a different readout when assayed systemically, for example through plasma. The inflammatory pathway is closely linked to the hemostatic pathway, and the upregulation of cytokines is able to trigger coagulation [99] and, as part of a feedback loop, coagulation is able to trigger inflammation [100,101], leading to a chronic immunothrombotic environment in CCM.

### 6.3. The Role of the Microbiome in Innate Immunity and Lesion Severity

Preclinical studies by Tang et al. show that the activation of the toll-like receptor-4 (TLR-4) pathway with lipopolysaccharide from gram-negative bacteria is able to propel the growth of lesions in mice resistant to the loss of CCM genes [102]. Importantly, Tang and collogues show that the administration of IL-1β and polyinosinic-polycytidylic acid (the ligand for TLR3) increased lesion burden suggesting a potential role for immune ligands in driving CCM [102]. Gram-negative *Bacteroides* (a potential source of lipopolysaccharide) are more prevalent in CCM patients compared to healthy controls [103]. These data suggest that circulating factors derived from the gut microbiome contribute to CCM disease.

## 7. The CCM Microenvironment

In CCM research, the heterogeneity of cavernomas has been largely underappreciated. Today we are beginning to understand that, like tumors, CCM lesions are heterogenous and have a distinct microenvironment. CCM lesions develop in endothelial cells of the CNS where they are part of the neurovascular unit (NVU). The NVU maintains the integrity of the blood-brain barrier (BBB) and consists of endothelial cells in close contact with the basement membrane, pericytes, neurons, microglia and astrocytic end-feet.

Gap junctions are essential for intercellular communication within the NVU, however, their role in CCM pathology is not fully understood. Using preclinical models of CCM, Johnson et al. showed that the gap junction, connexin 43, is upregulated in endothelial cells and pericytes [104]. The upregulation of connexin 43 in endothelial cells results in the mislocalization of zonula occludens-1 and claudin-5 [104], two tight junction proteins that regulate permeability at the BBB [105]. These data suggest that a high level of connexin 43 contributes to the increased permeability in CCM lesions. Intriguingly, the beta-blocker propranolol reduces lesion burden and cadaverine leakage in CCM-deficient mice [106]. In addition, propranolol also improves the interaction between endothelial cells, pericytes and astrocytes in these mice [106]. This suggests that the direct interactions within the NVU, mediated by gap junctions, also play an important role in modulating vascular stability in CCM.

Although CCM proteins are also expressed by other non-endothelial cells in the NVU (pyramidal neurons, astrocytes and mural cells) [107,108,109], CCM has largely been studied as an endothelial disease. The contribution of these other cell types to CCM pathology is now being revealed [110,111]. In a recent paper, Globisch et al. report an increased level of TF in the molecular and granular layers of the cerebellum [17]. They also show that TF colocalizes with glial fibrillary acidic protein, a marker of astrocytes. Astrocytes are known to be the major producers of TF in the brain [25]. Pericytes also express TF and are able to activate the extrinsic blood coagulation pathway [112]. These data indicate the role that non-endothelial cells in the NVU may have in modulating coagulation in CCM.

Further interplay between astrocytes and endothelial cells in CCM has been recently reported. As a consequence of increased KLF2/4, CCM-deficient endothelial cells increase their production of NO via eNOS, which in turn stabilizes hypoxia-inducible factor-1 alpha (HIF1α) in astrocytes, and leads to increased production of vascular endothelial growth factor (VEGF) [113]. Astrocyte-derived VEGF is then able to trigger angiogenic pathways and dysregulate intercellular junctions in endothelial cells [114]. Interestingly, thrombin can induce the phosphorylation and activation of eNOS in endothelial cells [115,116]. The increased coagulation in CCM results in cerebral hypoxia within the brain parenchyma [17], and this hypoxic environment allows for the stabilization of HIF1α and the subsequent induction of VEGF production by astrocytes. Overall, these studies suggest that there is a direct mechanism by which coagulation modulates astrocyte function in CCM (Figure 2).

Astrocytes, oligodendrocytes, neurons and endothelial cells express PARs which are activated by coagulant proteins including thrombin, activated FVIIa and FXa [117]. In vitro studies have shown that thrombin may be protective or detrimental to cells in the NVU based on the dose [117]. Low levels of thrombin stimulate the expression of PAI-1 in reactive astrocytes, protecting them and neurons from cell death while high levels of thrombin trigger the digestion of the extracellular matrix by metalloproteases and promote neuronal retraction and death [117]. In addition, in CCM, there is an increased proliferation of reactive astrocytes and microglia [12,118]. These cells also lose their ability to regulate glutamate and instead, increase the production of reactive oxygen species and inflammatory cytokines which are harmful to the surrounding cells and could potentially exacerbate patient symptoms [117].

Little is known about the direct effect of CCM on neurons other than the neurological symptoms that result from cavernomas. Globisch et al. reported hypoxia around the lesions of CCM-deficient mice, but they did not characterize all of the cell types undergoing hypoxic stress. However, it is quite plausible that hypoxia in the brain parenchyma dysregulates neuronal function, possibly severely enough to result in neuronal death which in turn may contribute to the neurological symptoms observed in patients with CCM. The CCM microenvironment is summarized in Figure 2, where the major topics discussed in this review (anticoagulation, thrombosis, hypoxia the pathophysiological mechanisms of endothelial cells and surrounding cells) are highlighted.

## 8. Anticoagulants for Patients with CCM

Due to the bleeding symptoms associated with CCM, an ongoing debate in the CCM field is whether or not anticoagulants are safe for CCM patients. For a long time, anticoagulants were thought to promote cerebral hemorrhage, however, recent data suggest that this may not be the case. In 2013, Flemming et al. reported that the use of antithrombotic agents does not actually promote hemorrhages in patients with intracerebral cavernous malformations [119]. Likewise, a case report describing the normalization and regression of low-flow venous malformation by antiplatelet treatment promotes the benefits of antithrombotic therapy in vascular malformations [120]. In 2019, Zuurbier et al. performed a population-based cohort study and meta-analysis and found that long- term antithrombotic therapy is associated with a lower risk of subsequent cerebral hemorrhage and/or focal neurological deficits [121]. In addition, another clinical study showed that aspirin combined with statin therapy significantly reduces the odds of acute hemorrhage in patients with CCM [122]. Interestingly, propranalol (under phase II clinical trial to treat CCM) has been reported to inhibit thrombin-induced platelet aggregation via pathways unrelated to its beta-adrenergic function [123,124]. It remains to be seen from the Treat_CCM clinical trial [125], if propranalol at the dose administered, would impact coagulation and intracranial bleeding in CCM patients. Taken together, these studies indicate that the use of anticoagulants does not promote hemorrhage in CCM patients and, in the long run, that such therapies could offer a protective effect against intracranial bleeding.

However, this is an area of research that has not been addressed extensively with mouse models, at least in part due to the limitations of using the standard mouse models as noted above. Newer mouse models have been developed in which the disease progression is more prolonged and the distribution of lesions is less restricted. Additionally, these models may more faithfully recreate the inflammatory and bleeding phenotypes seen in patients. Such models should be useful for studying the long-term efficacy of anticoagulant therapy in CCM [126,127,128,129].

## 9. Discussion

The hemostatic system is often overlooked when investigating vascular malformations. Therefore, deeper scientific investigations of hemostasis in CCM have the potential to establish a trove of new knowledge about this complex disease and enable better and more precise treatments. Recent studies describe that the hemostatic system in CCM is disrupted and promotes cerebral hemorrhage and blood clots that further obstruct oxygen delivery to the brain. These findings may provide the link between endothelial dysfunction and the neurological symptoms that CCM patients experience. These studies also raise the possibility that CCM, at least to some extent, can be termed a hemostatic disease. The fact that CCM patients receiving antithrombotic treatment for the prevention of cardiovascular disease, experience a reduced risk of hemorrhage demonstrates the impact of hemostasis in CCM.

Vascular heterogeneity of cerebral cavernomas contributes further to disease complexity and might complicate treatment. It is unknown if the loss of CCM genes simultaneously activates both pro and anticoagulant pathways or if the activation of one pathway triggers the compensatory activation of the other in an attempt to establish hemostatic balance. Although activated platelets are present in CCM pathology, the molecular interactions between these platelets and lesion endothelial cells have yet to be established. Still, activated platelets pull on fibrin threads, leading to increased clot contraction and the formation of polyhedrocytes, elements implicated in stable clot formation in CCM.

When viewing CCM as a disease characterized by excessive bleeding, the existence of stable clots in vivo might seem counterintuitive, and it is not yet known why the clotting process does not sufficiently stem hemorrhage. A better understanding of the effects that astrocytes and other non-endothelial cells have on hemostasis in CCM would also help determine the most optimal therapeutic approach. Likewise, ongoing efforts to find relevant circulating biomarkers to accurately detect a patient’s specific and unique disease status should ideally deliver a hemostatic and inflammatory profile of the patient to enable a personalized treatment strategy. Transcriptomic and proteomic analyses on CCM patient material, in combination with new mouse models that better recapitulate the human disease, represent a promising approach for establishing precision medicine to restore the hemostatic balance in CCM pathology.

## Figures and Tables

**Figure 1 ijms-23-12575-f001:**
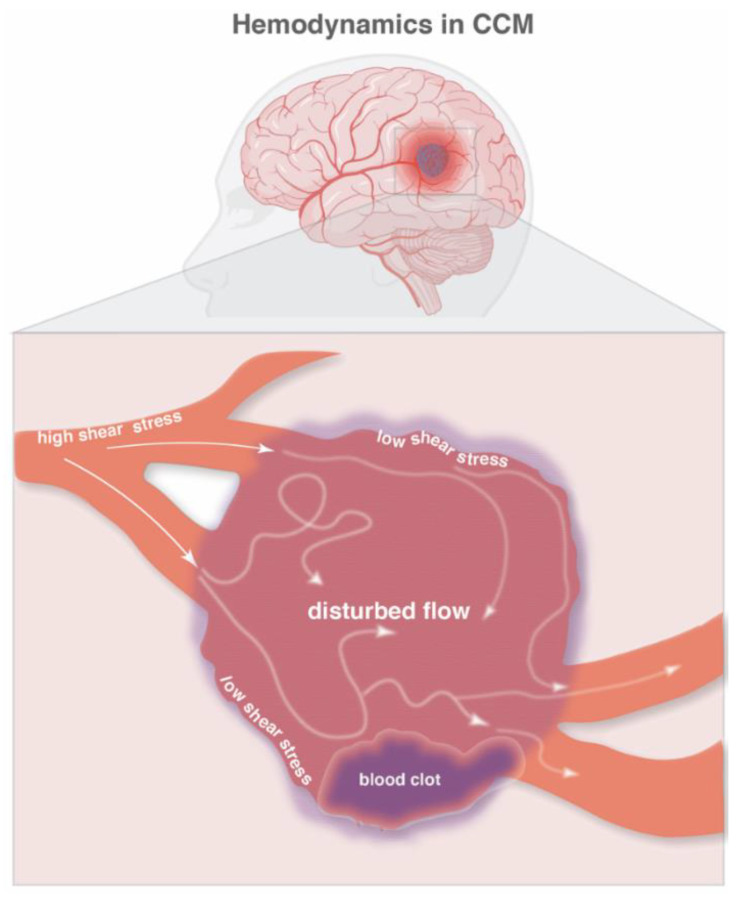
**Hemodynamics in CCM**. A schematic representation of blood flow in a cavernoma. Normal blood flow (solid white arrows) results in high shear stress on endothelial cells while disturbed blood flow (diffused white arrows) results in low shear stress on endothelial cells. A blood clot (dark purple) may further disturb the already reduced blood flow in the cavernoma. In addition, a blood clot may also obstruct nearby blood vessels and further reduce blood flow. The figure was assembled with Adobe Illustrator and BioRender (agreement number QG24D9MLOD).

**Figure 2 ijms-23-12575-f002:**
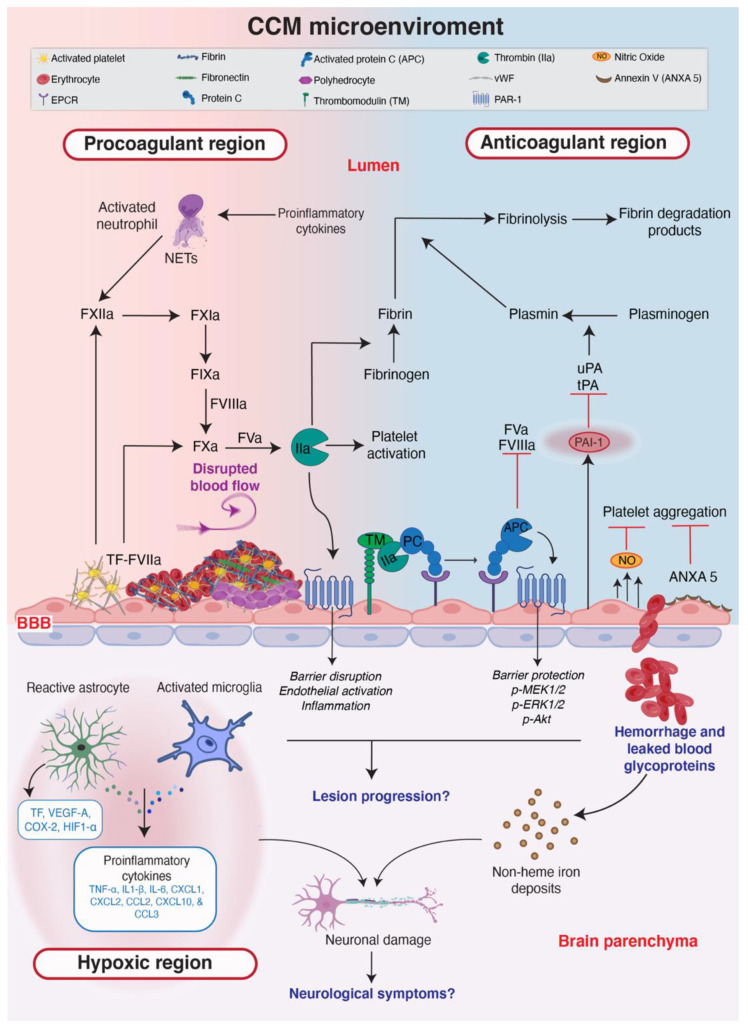
**The CCM microenvironment.** A schematic representation of the CCM microenvironment of procoagulant (red) and anticoagulant (blue) regions, where endothelial cells are the key regulators of hemostasis, neuroinflammation and bleeding. Endothelial cells interact with blood cells and glycoproteins to regulate coagulation and anticoagulation. CCM-deficient endothelial cells cannot maintain a hemostatic balance and this results in neuroinflammation and hypoxia in the brain parenchyma (light purple). Neuroinflammation, hypoxia and bleeding are all hallmarks of CCM. A simplified version of the coagulation cascade is shown in the lumen and clinical aspects of cavernomas are written in dark blue. Tissue Factor (TF); Urokinase plasminogen activator (uPA); Tissue plasminogen activator (tPA); Neutrophil extracellular traps (NETs); Blood-brain barrier (BBB). The figure was assembled with Adobe Illustrator and BioRender (agreement number QG24D9MLOD).

**Table 1 ijms-23-12575-t001:** Blood glycoproteins related to the coagulation cascade identified in CCM.

CCM Article	Murine Blood Glycoproteins	Shown in Humans?
Lopez-Ramirez et al., 2019 [12]	APC	N
Yau et al., 2022 [16]	Fibrinogen/fibrin	N
Globisch et al., 2022 [17]	vWF	Y
	Fibrinogen/fibrin	Y
	Fibronectin	N

APC = activated protein C; vWF = von Willebrand factor; Y = Yes; N = No.

**Table 2 ijms-23-12575-t002:** Blood cells identified in CCM.

CCM Article	Murine Blood Cells	Shown in Humans
Shi et al., 2009 [49]	N/A	Macrophages T-cells B-cells
Shi et al., 2016 [50]	B-cells	N/A
Yau et al., 2022 [16]	Macrophages	N
	Neutrophils T-cells B-cells Platelets	Y N N N
Globisch et al., 2022 [17]	Activated platelets	N
	Polyhedrocytes	Y

Y = Yes; N = No; N/A = not available.

**Table 3 ijms-23-12575-t003:** Cytokine and chemokines in CCM associated with lesion severity and symptoms.

CCM Article	In Situ Brain Cytokines/Chemokines	Circulating Cytokines/Chemokines	Associated Symptoms
Noshiro et al., 2012 [37] (Clinical; mRNA)	↑ IL-6	N/A	Increased number of hemorrhage events Increased occurrence of bleeding in brain stem
Girard et al., 2018 [98] (Clinical)	N/A	↑ MMP2 ↑ ICAM-1 ↓ MMP9	Increased seizures
		↓ VEGF ↓ Endoglin	Recent bleeding
		IL-2 Interferon gamma TNFα IL-1β	Increased incidence of events (bleeds/lesional growth)
		↓ IL-10 ↓ CCL2/MCP1 ↓ ROBO4	Iron deposition from bleeding lesions
Girard et al., 2018 [96] (Clinical)	N/A	↓ CD14 ↓ IL-6 ↓ VEGF ↑ IL-1β ↑ sROBO4	Symptomatic hemorrhagic expansion
Lyne et al., 2019 [97] (Clinical)	N/A	sCD14 VEGF IL-10 CRP sROBO4	Bleeding
Yau et al., 2022 [16] (Preclinical)	IL-1β TNF CXCL1/KC/GRO CXCL2/MIP-2 CCL2/MCP-1 CXCL10/IP-10 IL-6 CCL3/MIP-1α	N/A	Lesion severity

↑ = increase; ↓ = decrease; N/A = not available.

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
