# Peer review of "Dysregulated Hemostasis and Immunothrombosis in Cerebral Cavernous Malformations"

_ijms, 2022, doi:10.3390/ijms232012575_

Round 1
Reviewer 1 Report
This review attempts to show that hemostatic dysregulation and immunothrombosis are involved in cerebral cavernous malformation (CCM) disease, and argues for anticoagulant therapy for patients with CCM disease. Unfortunately, the pre-clinical evidence for this idea utilized acute models with animals harvested at a very young age. Notably, the much cited Reference #17 (Globisch et al. 2022) describes studies in which mice, in an acute model, were harvested between P6 and P9. Moreover, in many of these acute animal models, the CCM lesions are restricted to the cerebellum, which is not the case in human patients with CCM disease, whose lesions are spread throughout the entire brain. The present review should discuss the limitations in using the acute animal models in understanding human CCM disease, and, in particular, hemorrhage from the CCM lesions in adult patients. Although this is suggested in the last sentence in the Discussion, it would be helpful if the authors would list which features of “new mouse models that better recapitulate the human disease” would be useful in better understanding hemostatic dysregulation and immunothrombosis in human CCM disease.
Minor comments:
Figure 1 is not indicated in the manuscript text.
Reference numbers are included in Table 1, but not in Tables 2 and 3.
Author Response
* The present review should discuss the limitations in using the acute models in understanding human CCM disease, and, in particular hemorrhage from the CCM lesions in adult patients. Although this is suggested in the last sentence in the Discussion, it would be helpful if the authors would list which features of “new mouse models that better recapitulate the human disease” would be useful in better understanding hemostatic dysregulation and immunothrombosis in human CCM disease.
Reply: We have now included text which more clearly acknowledges these limitations. We have also included text which explicitly identifies the features of newer mouse models that will likely benefit the study of hemostasis in CCM, namely the slower disease progression, the wider distribution of CCM lesions within the CNS, and the inflammatory and bleeding profile of the lesions that arise.
Please find the new text on page 5, line 168-176 and on page 17, line 548-554.
*Figure 1 is not indicated in the manuscript text. Reference numbers are included in Table 1, but not in Tables 2 and 3.
Reply: Please notice that we have included one additional figure in the review (Figure 1, Hemodynamics in CCM, first presented in section 4) and the earlier figure 1 has now become figure 2 and is presented within the text of section 7. We have also updated tables 2 and 3 to include the references numbers, as in table 1. The tables are also positioned in the text when first mentioned.

Reviewer 2 Report
The review of Globisch and coauthors is devoted to the discussion of the role of the hemostasis system and the phenomenon of thromboinflammation in the occurrence, development and complications of cerebral cavernous malformations (CCM). The review is well-written and well-structured and can be accepted for publication in the IJMS in its present form. However, it seems to me that adding/changing the following points may make it easier for readers to understand the material.
Minor points.
1. The authors have just published a research paper (in Blood Journal, BLD-2021-015350R1) on the topic of this review. It seems to me that the article would benefit from an explicit mention of this article at the beginning and in the Abstract.
2. In section 3, the authors discuss the involvement of the contact pathway of blood coagulation in the development of CCM and its complications. However, the involvement of the contact pathway in thrombosis is already debatable, so I would advise the authors to soften the following statements. A. “the intrinsic pathway <…> when in contact with a negatively charged surface, trigger the blood coagulation cascade” – not only negatively charged surfaces and not every negatively charged surface induces blood coagulation. B. “polyphosphates[33] which are known to activate circulating factor XII”- it has been shown in multiple studies, that originated from platelets polyphosphates do not activate FXII. C. “platelets which in turn activate factor XII” – again, has been shown in multiple studies, that only coated platelets activate FXII and VWF does not induce coated platelet formation.
3. There is a tipo in Section 3.1: “Tissue factor <…> when in contact with blood, it strongly binds circulating FVII to create a complex and triggers a catalytic event that results in the activation of FVII (FVIIa)”
4. The authors name Section 3.2. as describing the intrinsic pathway, but they discuss mainly VWF, which is not part of the intrinsic pathway.
5. Section 3.4. “CCM lesions have stable thrombi with polyhedrocytes” is Section 4 is very intriguing and I was very interested in this topic as a result of reading it. However, in my opinion, the material in this section is not structured enough and I would like to have clearer answers to the following questions. In which conditions the polyhedrocytes were observed in CCM? Whether their presence correlated with the size of the cavity, the presence of other markers, or prognosis? Please provide a clearer reasoning on the topic of their formation as a result of clot retraction. It is well known that there are RBCs in venous thrombi, since there are slow flows and the plasma part of hemostasis predominates, while there are no RBCs in arterial thrombi, since fast blood flows lead to the accumulation of RBCs closer to the center of the vessel and "pushing" platelets to the periphery. The discussion of how the flow behaves in the cavity is very important here.
3. There are not enough illustrations for the material presented in Section 4. A small drawing showing how blood stasis and turbulent flows occur in the CCM, and how they change during blood clotting, will greatly facilitate the perception of hemodynamics.
4. It is advisable to add a couple of words about PAR-2 on endothelial cells (doi: 10.1016/j.thromres.2019.01.009) to Section 6.1
Author Response
- The authors have just published a research paper (in Blood Journal, BLD-2021-015350R1) on the topic of this review. It seems to me that the article would benefit from an explicit mention of this article at the beginning and in the Abstract.
Reply: We have added text to the abstract (page 1, line 13-14) and to the introduction (page 2, line 62) that makes clear that our lab has recently published on the topic of hemostasis in CCM which we hope will add context and provide the audience with an explanation for why we are extensively citing our recent studies.
- In section 3, the authors discuss the involvement of the contact pathway of blood coagulation in the development of CCM and its complications. However, the involvement of the contact pathway in thrombosis is already debatable, so I would advise the authors to soften the following statements. A. “the intrinsic pathway <…> when in contact with a negatively charged surface, trigger the blood coagulation cascade” – not only negatively charged surfaces and not every negatively charged surface induces blood coagulation. B. “polyphosphates[33] which are known to activate circulating factor XII”- it has been shown in multiple studies, that originated from platelets polyphosphates do not activate FXII. C. “platelets which in turn activate factor XII” – again, has been shown in multiple studies, that only coated platelets activate FXII and VWF does not induce coated platelet formation.
Reply: We thank the reviewer for highlighting this matter and to address the concerns described in point 2, we have rephrased the text in section 3.1 “The Hemostatic system” on page 3, line 107-108 and in section 3.2 “Are vWF-activated platelets the activation of the intrinsic pathway in CCM?” on page 4, line 142-153. This to give important context of relevance in the contact pathway and coagulation. We have also updated the section 3.2 with new references (ref no. 34-36).
- There is a tipo in Section 3.1: “Tissue factor <…> when in contact with blood, it strongly binds circulating FVII to create a complex and triggers a catalytic event that results in the activation of FVII (FVIIa)”
Reply: We appreciate the careful reading and have corrected the typographical error raised in point 3, on page 3, line 112: “Tissue factor <…> when in contact with blood, it strongly binds circulating FVII to create a complex and triggers a catalytic event that results in the formation of activated FVII (FVIIa).
- The authors name Section 3.2. as describing the intrinsic pathway, but they discuss mainly VWF, which is not part of the intrinsic pathway.
Reply: For point 4, we thank the reviewer for pointing this out and as described in point 2, we have now adjusted the headline of section 3.2 “Is the intrinsic pathway activated during blood coagulation in CCM? to “Are vWF-activated platelets supporting the activation of the intrinsic pathway in CCM?” that we now hope describes the topic of the section better.
- Section 3.4. “CCM lesions have stable thrombi with polyhedrocytes” is Section 4 is very intriguing and I was very interested in this topic as a result of reading it. However, in my opinion, the material in this section is not structured enough and I would like to have clearer answers to the following questions. In which conditions the polyhedrocytes were observed in CCM? Whether their presence correlated with the size of the cavity, the presence of other markers, or prognosis? Please provide a clearer reasoning on the topic of their formation as a result of clot retraction. It is well known that there are RBCs in venous thrombi, since there are slow flows and the plasma part of hemostasis predominates, while there are no RBCs in arterial thrombi, since fast blood flows lead to the accumulation of RBCs closer to the center of the vessel and "pushing" platelets to the periphery. The discussion of how the flow behaves in the cavity is very important here.
Reply: With point 5, we have done our best to restructure the section 3.4 “CCM lesions have stable thrombi with polyhedrocytes”, and hope that our revised text is better able to answer the questions posed by the reviewer (page 5, line 181-184, page 5-6, line 192-205). We think that some of these questions are too specific for the scope of this review article, but gladly refer back to our recent paper which we believe to be the first to identify polyhedrocytes within CCM lesions. Also, we think it is important to acknowledge that while RBCs are more associated with venous thrombi, they are also observed in arterial thrombi as well as shown by Chernysh and colleagues (doi: ARTN 511210.1038/s41598-020-59526-x). We have expanded section 3.4 to discuss the behavior of flow within the lesion lumen (including an additional schematic figure).
- There are not enough illustrations for the material presented in Section 4. A small drawing showing how blood stasis and turbulent flows occur in the CCM, and how they change during blood clotting, will greatly facilitate the perception of hemodynamics.
Reply: In regards to the next point, we have constructed a new illustration, supplementing the previous figure. The newly created figure 1, illustrates blood stasis and flow and is presented in section 4. Figure 2 (earlier Figure 1), the overview of the microenvironment in CCM is presented in section 7.
- It is advisable to add a couple of words about PAR-2 on endothelial cells (doi: 10.1016/j.thromres.2019.01.009) to section 6.1.
Reply: In reference to the final point, we have taken the advice and have expanded our section on PARs to include information about PAR-2 in endothelial cells. Please find the new text on page 11, line 386-395.
In closing, we once again thank the reviewers and the editors for their efforts, and we hope that our resubmission is acceptable for publications.
